# Automatic Extraction of Marine Aquaculture Zones from Optical Satellite Images by R$^3$Det with Piecewise Linear Stretching

**Yujie Ma** [1,†] ![ID], **Xiaoyu Qu** [2,†], **Cixian Yu** [1], **Lianhui Wu** [3] ![ID], **Peng Zhang** [1], **Hengda Huang** [1], **Fukun Gui** [4] **and Dejun Feng** [4,*]

1  Marine Science and Technology College, Zhejiang Ocean University, Zhoushan 316000, China
2  School of Fishery, Zhejiang Ocean University, Zhoushan 316000, China
3  Department of Marine Resources and Energy, Tokyo University of Marine Science and Technology, Tokyo 108-8477, Japan
4  National Engineering Research Center for Marine Aquaculture, Zhejiang Ocean University, Zhoushan 316000, China
*  Correspondence: fengdj@zjou.edu.cn; Tel.: +86-152-5708-0767
†  These authors contributed equally to this work.

**Abstract:** In recent years, the development of China's marine aquaculture has brought serious challenges to the marine ecological environment. Therefore, it is significant to classify and extract the aquaculture zone and spatial distribution in order to provide a reference for aquaculture management. However, considering the complex marine aquaculture environment, it is difficult for traditional remote sensing technology and deep learning to achieve a breakthrough in the extraction of large-scale aquaculture zones so far. This study proposes a method based on the combination of piecewise linear stretching and R$^3$Det to classify and extract raft aquaculture and cage aquaculture zones. The grayscale value is changed by piecewise linear stretching to reduce the influence of complex aquaculture backgrounds on the extraction accuracy, to effectively highlight the appearance characteristics of the aquaculture zone, and to improve the image contrast. On this basis, the aquaculture zone is classified and extracted by R$^3$Det. Taking the aquaculture zone of Sansha Bay as the research object, the experimental results showed that the accuracy of R$^3$Det in extracting the number of raft aquaculture and cage aquaculture zones was 98.91% and 97.21%, respectively, and the extraction precision of the area of the aquaculture zone reached 92.08%. The proposed method can classify and extract large-scale marine aquaculture zones more simply and efficiently than common remote sensing techniques.

**Keywords:** image stretching; R$^3$Det; aquaculture; remote sensing; deep learning

## 1. Introduction

In recent years, with the continuous increase in fishing intensity, marine aquatic resources have gradually decreased. To meet the demand for aquatic products, the marine aquaculture industry in various countries has been extensively developed [1,2]. According to the statistics of global aquaculture production in 2022, the total amount of aquaculture in Asia accounts for 91.61% of the global total. China accounts for 62.77% of the total aquaculture in Asia, and the aquaculture industry has an important position in China [3]. As an important part of the aquaculture industry, the rapid expansion of marine aquaculture has brought great challenges to the marine ecological environment [4]. The geographical and environmental conditions of marine aquaculture zones weaken the exchange capacity of internal and external water bodies. Meanwhile, the pollutants such as fish excrement, residual bait, and antibiotics exceeded the environmental carrying capacity, causing serious water pollution in the marine [5–7]. Moreover, the harsh natural environment such as

typhoons is constantly threatening the development of marine aquaculture. The huge waves caused by typhoons can bring devastating blows to the aquaculture zones and cause incalculable economic losses [8,9]. Reasonable planning of aquaculture zones, control of aquaculture scale, and reduction in aquaculture density can reduce aquaculture risks and improve economic efficiency. Therefore, it is significant to accurately obtain the spatial distribution, aquaculture quantity, and the area of the marine aquaculture zone [10].

Marine aquaculture zones are widely distributed, numerous, and complex in the environment, which makes it difficult to obtain accurate information. The traditional method of manually determining the number and area of aquaculture zones is time-consuming and labor-intensive. High-resolution remote sensing satellite images are featured with a wide imaging range and high imaging accuracy; thus, they have obvious advantages for the extraction of large-scale and small target objects and have been widely developed in the extraction of marine aquaculture zones [11–13]. Jayanthi adopted visual interpretation to extract aquaculture zones along the southeastern coast of India and statistically analyzed changes in aquaculture zones [14]. Seto and Fragkias adopted visual interpretation to extract information on aquaculture zones in QuickBird remote sensing images to investigate the impact of the Ramsar Convention on Wetlands on the aquaculture industry. It was found that the implementation of the Ramsar Convention on Wetlands did not slow down the development of aquaculture in Ramsar wetlands [15]. Although the visual interpretation method has higher extraction accuracy, the workload is high in terms of time consumption, and the extraction accuracy depends on the experience of the interpreter. This method has strong subjectivity and is not suitable for the extraction and quantitative analysis of large-scale aquaculture zones. To further improve the extraction accuracy and efficiency of aquaculture zones, experts and scholars proposed methods such as information extraction based on spatial structure [16,17], information extraction based on ratio index analysis [18], information extraction based on correspondence analysis [19], and object-oriented information extraction [20–22]. The aquaculture zone is effectively extracted by classifying the spatial, spectral, texture, and shape features of the object. Although the use of traditional remote sensing technology can achieve good results in the extraction of a single aquaculture type in a small range, with the expansion of the aquaculture range, the aquaculture environment becomes increasingly complex. Meanwhile, the traditional extraction method is affected by factors such as "salt-and-pepper noise", "same substance with different spectrum", and "same spectrum foreign matter", which lead to a decrease in the extraction accuracy.

In recent years, deep learning has achieved great success in the field of computer vision. Because of its generalization and robustness, it has been gradually applied to aquaculture extraction [23,24]. In the face of high-density, large-scale marine aquaculture zones and aquaculture sea conditions with complex spectral information, deep learning has better feature analysis capabilities and can achieve better extraction accuracy. For example, Cui improved the U-Net network structure by adding a pyramid up-sampling module and a squeeze-excitation module (PSE), which solved the problem of fuzzy boundaries. The network was applied to extract the raft aquaculture zone in the east of Lianyungang, China [25]. Liu et al. proposed a multisource feature fusion target extraction method based on DeepLabv3, which could effectively extract marine aquaculture zones with weak signals [26]. Fu et al. proposed a hierarchical cascade convolutional neural network (HC-Net), which could effectively extract multiscale information from images and map marine aquaculture zones more finely [27]. On the basis of Sentinel-2 multispectral scan imaging (MSI) image data, the improved U-Net model reduces the edge-sticking phenomenon and improves the extraction accuracy of the aquaculture zone [28]. However, to improve the extraction accuracy of aquaculture zones, most of the existing research took raft aquaculture in a specific zone as the research object, and the extraction of aquaculture areas in this area was realized by improving the network structure. This method has high professional requirements for scholars, and the extraction range is limited, which makes it difficult to be applied to statistics and monitoring in large-scale aquaculture zones. In addition, the

marine aquaculture management department usually needs to conduct statistical analysis and monitoring of a variety of aquaculture types (mainly including rafts and cages), and the extraction of a single type of aquaculture zone cannot provide substantial help for the statistical management of aquaculture zones.

As an improved single-stage detector, $R^3$Det has higher extraction speed and extraction accuracy, and its rotation bounding box has a higher fitting effect with the extraction target [29]. In a previous study, Ma et al. applied it to the extraction of cage aquaculture zones in Fujian Province, and good results were achieved in the extraction of large-scale single-type aquaculture zones [30]. However, the extraction of cage aquaculture zones is still affected by similar features due to the complex marine environment. For example, some raft aquaculture and cage aquaculture zones have similar characteristics, which reduces the extraction accuracy of the model for the aquaculture zone. Meanwhile, the statistics of a single type of cage aquaculture do not result in practical effects on the management of marine aquaculture. In this case, it is crucial to reduce the influence of the aquaculture background on the aquaculture zone and realize the classification and extraction of different aquaculture types. Moreover, further improving the extraction accuracy in a simple way needs to be investigated.

As a high-efficiency and low-cost image processing technology, image enhancement can highlight the important details according to qualitative criteria so as to improve the extraction of the target [31–34]. In particular, the method of enhancing image contrast using an image histogram has gradually been accepted because of its ability to process images more adaptively [35,36]. Therefore, this study proposes a combination of histogram-based piecewise linear stretching and $R^3$Det to extract marine aquaculture zones. The image of the offshore aquaculture zone is enhanced by piecewise linear stretching, and then the aquaculture zone is classified and extracted by $R^3$Det. It was found that piecewise linear stretching can effectively suppress the grayscale range of raft aquaculture and cage aquaculture zones and reduce the color difference in the raft aquaculture zone due to different aquaculture periods. Meanwhile, it can improve the contrast of the image and reduce the influence of the aquaculture background on the extraction accuracy of the aquaculture zone. The experimental results indicate that the proposed method is simple and efficient to improve the classification and extraction accuracy of offshore aquaculture zones.

## 2. Study Area and Data

### 2.1. Study Area

Sansha Bay is in the northeastern part of Fujian Province (26°30′–26°58′N, 119°26′–120°10′E) (Figure 1). It is a semi-closed world-class natural deep-water port composed of Dongchong Peninsula and Jianjiang Peninsula, with a water area of 714 square kilometers [37,38]. Moreover, Sansha Bay is an important aquaculture zone in China, where cage aquaculture and raft aquaculture are the two main aquaculture types (Figure 2). Traditional aquaculture cages are mainly used for fish farming and are composed of rigid frames (wood or steel structure), flexible nets, floats (EPS floating balls), and anchors. The cages are always floating on the water surface and can be seen as a gray-white color in the remote sensing images (Figure 2c) due to the cage frame and floats. Raft aquaculture mainly uses floats and ropes to form floating rafts, which are fixed to the seabed with cables so that the seedlings of seaweed and sessile animals (such as mussels) are fixed on the slings suspended on the floating raft. The raft aquaculture zone has a dark-gray band on the image (Figure 2d), and the color tone of a single aquaculture zone is uniform. Meanwhile, the depth of tone varies between different raft aquaculture zones at different aquaculture stages. In addition, there are many estuaries and islands in Sansha Bay, and the sand in the near-coastal zone is accumulated, resulting in a complex sea environment.

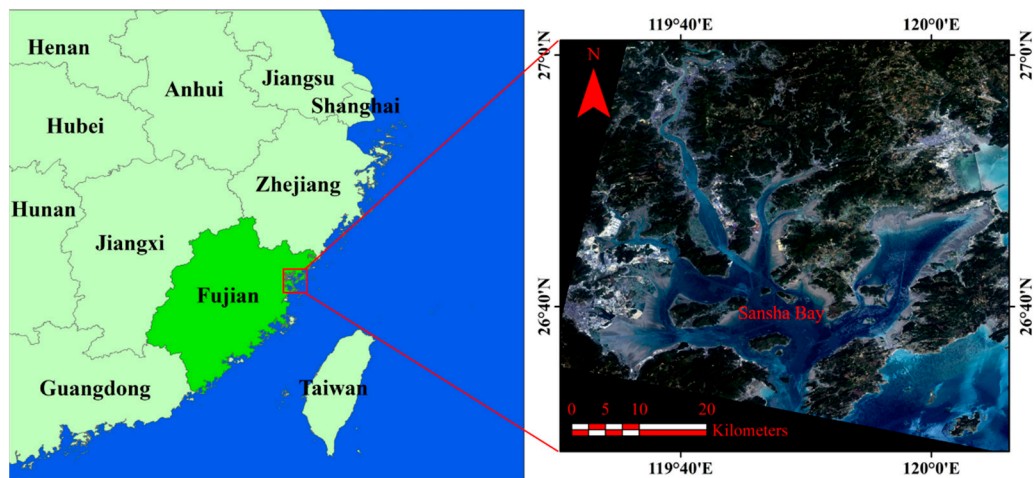

**Figure 1.** Location of the study area of Sansha Bay, Fujian Province.

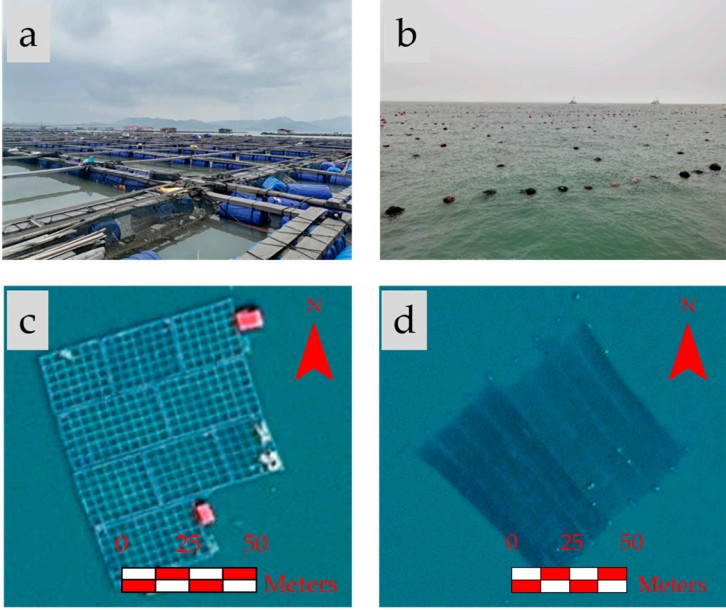

**Figure 2.** Photography of cage (**a**) and raft (**b**). Level 19 remote sensing image of cage (**c**) and raft (**d**) in Google Earth.

### 2.2. Data and Preprocessing

Optical remote sensing images can provide spectral information and capture ground objects with different spectral characteristics. Gaofen-6 (GF-6) is a low-orbit optical remote sensing satellite equipped with a multispectral high-resolution sensor PMS (panchromatic band spatial resolution of 2 m, multispectral band spatial resolution of 8 m) and a multispectral medium-resolution wide-width sensor WFV (multispectral band spatial resolution of 16 m), which can achieve a global observation, and the image data effectively cover the coastal areas of China; the specific parameters of the two sensors are shown in Table 1. Compared with Gaofen-1, Ziyuan-3, and other Gaofen series satellites, GF-6 PMS observation has a width of 95 km, and it has the advantages of large coverage, which help to avoid the influence of ground object reflectivity due to different image shooting times. In addition, GF-6 and GF-1 have multispectral high-resolution sensors with the same spatial resolution, enabling 2/8 m sensors to revisit the world in 1 day.

**Table 1.** Indicators of GF-6 satellite payload and performance.

| Sensor Type | | Spectral Range (nm) | Spatial Resolution (m) | Swath Width (km) | Revisit Period (Day) | Coverage Period (Day) |
|---|---|---|---|---|---|---|
| PMS sensor | Panchromatic | Panchromatic: 450–900 | 2 | 95 | 4 | 41 |
| | Multispectral | Blue: 450–520 Green: 520–590 Red: 630–690 NIR: 770–890 | 8 | | | |
| WFV sensor | Multispectral | Blue: 450–520 Green: 520–590 Red: 630–690 | 16 | 860 | 4 | 41 |

The raft aquaculture zone has different characteristics in different aquaculture stages. The zone with a longer aquaculture period is more distinct from the seawater background, while the characteristics of the cage aquaculture zone do not change with the aquaculture stage. To ensure that the raft aquaculture zone had obvious characteristics in the remote sensing images and to avoid the influence of cloud coverage, this study selected the panchromatic and multispectral images of PMS (17 April 2020). All images were preprocessed by ENVI5.3 software, in which panchromatic images were processed by radiometric calibration and orthorectification, and multispectral images were subjected to radiometric calibration, atmospheric correction, and orthorectification. In this way, the effects of unfavorable factors such as sensors and the atmosphere could be eliminated [39,40]. Meanwhile, to ensure the visual effect during data processing, it was necessary to upscale the image resolution to 2 m with ENVI5.3.

## 3. Research Methods

### 3.1. Extraction Process from Satellite Images

The operation process of extracting aquaculture zones in this study consisted of the following three stages: image processing, model training, and results analysis (Figure 3). In the first stage, the effects of factors such as sensors and the atmosphere were eliminated by image preprocessing and by constructing the NDWI (normalized difference water index, which can efficiently achieve the separation of water and land when faced with a large area of sea) model to realize the separation of water and land for the fused images, thus eliminating the influence of inland features on the extraction accuracy of aquaculture zones. However, it was found that some of the cages would be rejected as non-water bodies in the experiment because the aquaculture cages made of materials such as wooden boards or steel structures were floating on the sea surface, and the spectral reflectance was different from that of water bodies. Therefore, it was necessary to repair this part of the image information to ensure the integrity of the information to be identified, and then the image was stretched by piecewise linear stretching. In addition, in subsequent experiments, only the true-color images composed of three bands of red, green, and blue were used as data sources. In the second stage, to ensure the credibility of the experiment, it was necessary to divide the research area into a training set and a test set and expand the training samples for model training. In the third stage, the test set was input into the trained model for testing, the resulting image was obtained, and the accuracy evaluation and comparative analysis of the resulting images under different conditions were performed.

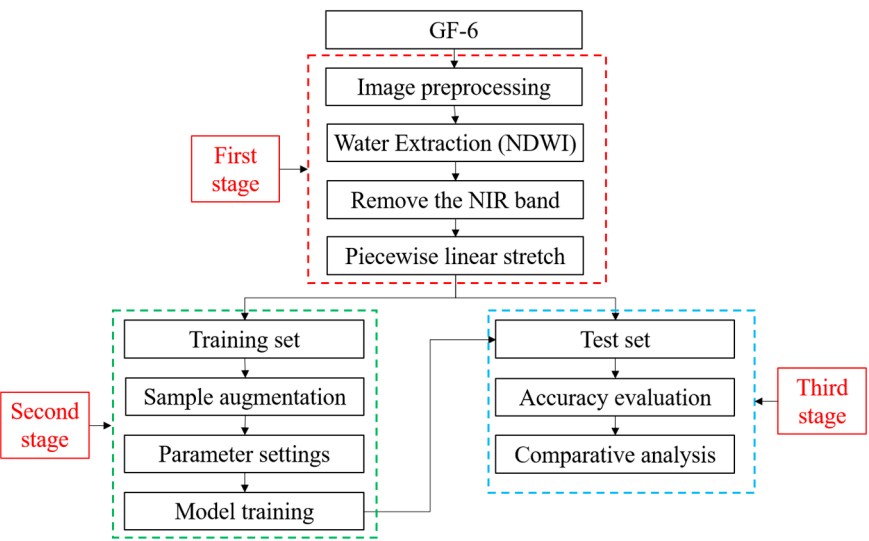

**Figure 3.** Flowchart of aquaculture zone extraction in this study.

### 3.2. Piecewise Linear Stretching Based on Histogram

An image histogram reflects the grayscale value distribution that represents the occurrence frequency of each grayscale value [41]. The piecewise linear stretching based on a histogram can highlight the region of interest by changing the grayscale value of the image pixel, enhancing the image contrast, and improving the image quality [42,43]. Sansha Bay aquaculture zones were mainly divided into three types: cage aquaculture zone, raft aquaculture zone, and non-aquaculture zone. Among them, the raft aquaculture zone had weak texture characteristics, and there were differences in aquaculture periods and zones. Although cage aquaculture had strong texture characteristics, the differences between some cages and raft aquaculture zones were small. To intuitively reflect the grayscale characteristics of three different aquaculture zones, we statistically evaluated the grayscale values of the different types of aquaculture zones in Figure 4, and the results are shown in Table 2. In the green band and blue band, the cage aquaculture zone had the largest average grayscale value, followed by the non-aquaculture zone, while the raft aquaculture zone had the smallest average grayscale value due to the influence of underwater aquaculture species; in the red band, the average grayscale value of the raft aquaculture zone was higher than that of the non-aquaculture zone. Therefore, to better reduce the complex appearance characteristics of the raft aquaculture zone and the influence of the aquaculture background, the effect of image recognition was improved by enhancing the contrast of the image. In this study, when adjusting the piecewise transformation points of piecewise linear stretching, the transformation points were set between the average grayscale value of the raft aquaculture zone and the non-aquaculture zone, as well as between that of the non-aquaculture zone and the cage aquaculture zone. The schematic of piecewise linear stretching is shown in Figure 5, and the corresponding calculation is shown in Equation (1).

$$g(y) = \begin{cases} \frac{c}{a} \times f(x) & 0 \leq f(x) < a \\ \left[\frac{d-c}{b-a}\right] \times f(x) + c & a \leq f(x) \leq b \\ \frac{255-b}{255-d} \times f(x) + d & b < f(x) \leq 255 \end{cases} . \tag{1}$$

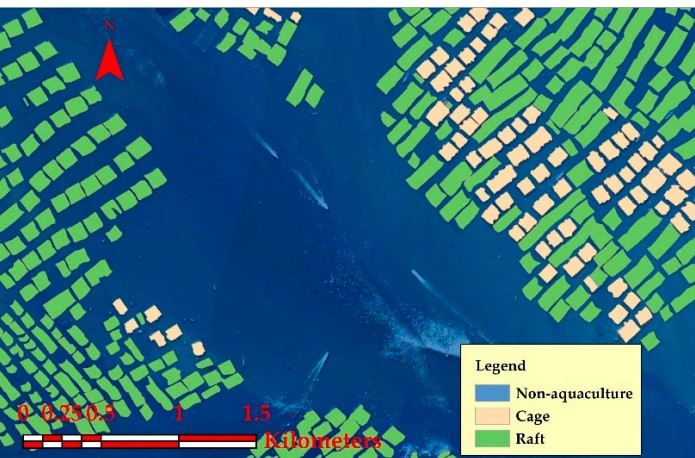

**Figure 4.** Thematic map showing the marine aquaculture zones in Sansha Bay.

**Table 2.** Average grayscale values of different types of aquaculture zones in the region shown in Figure 4 under different wavelength bands.

|  | Value (Red) | Value (Green) | Value (Blue) |
|---|---|---|---|
| Cage | 195.76 | 157.62 | 150.18 |
| Raft | 32.40 | 22.45 | 27.39 |
| Non-aquaculture | 12 | 72 | 70 |

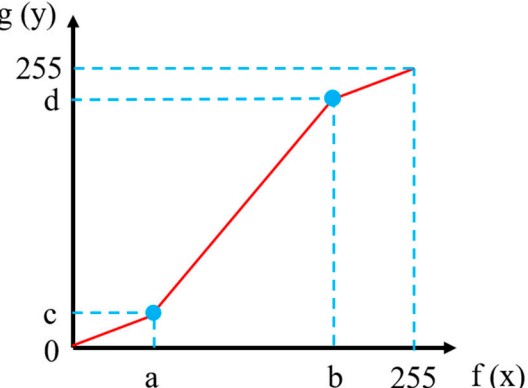

**Figure 5.** The schematic diagram of piecewise linear stretching. Here, f(x) represents the grayscale value of the original image; a and b represent the segment transformation points in the original image; g(y) represents the grayscale value after image enhancement; c and d represent the segment point after enhancement.

To compress the grayscale range of raft aquaculture and cage aquaculture, according to the actual stretching effect, the value of c should be within [0, a), and the value of d should be within (b, 255]. The images before and after piecewise linear stretching are shown in Figure 6. From the comparison of the enlarged area in the lower right corner of the two figures, it can be seen that, in the image after piecewise linear stretching, the grayscale interval of the raft aquaculture zone was obviously compressed, and the complex appearance characteristics caused by the aquaculture cycle were reduced, while the grayscale of the aquaculture cage was significantly improved.

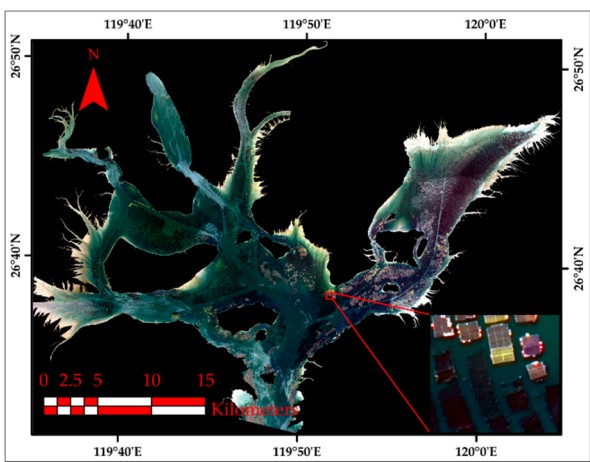
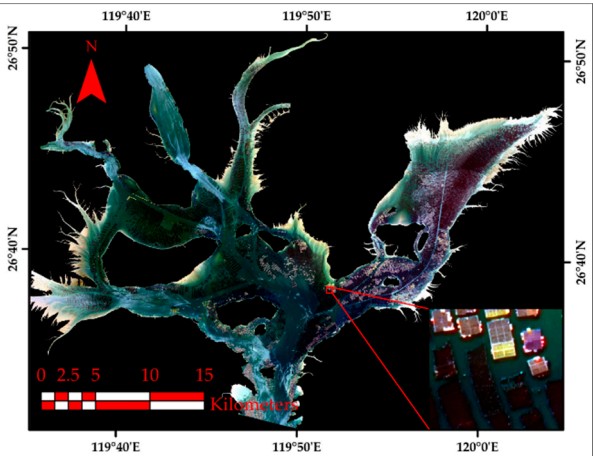

**Figure 6.** Image before (**left**) and after (**right**) piecewise linear stretching.

### 3.3. Dataset

In this study, the aquaculture zone of Sansha Bay was used as the dataset and divided into a training set and a test set. The training set was used to train the parameters in the model, and the test set was used to evaluate the generalization ability of the model. The traditional dataset division is mainly based on the training set to ensure the training effect of the model. However, aquaculture zones have different characteristics such as color and texture. To ensure the credibility of the model evaluation results, the test samples should contain as many characteristics of the identification target as possible. Therefore, the ratio of the training set and test set was adjusted to 0.4:0.6 by taking the aquaculture zone of Sansha Bay as the unit in this study [44–46]; the division results are shown in Figure 7a. To ensure the quality of training samples and alleviate the impact of the shortage of training samples on model training, this study used python to divide the training image into 135 images of 800 × 800 pixels as the data source (Figure 7b), and the labelme software was adopted to create a training set (Figure 7c). Then, the training set was augmented using data augmentation methods, including image translation, image flipping (horizontal, vertical, diagonal), and image brightness adjustment. Through data augmentation, the original 135 images were expanded tenfold, yielding 1350 training samples.

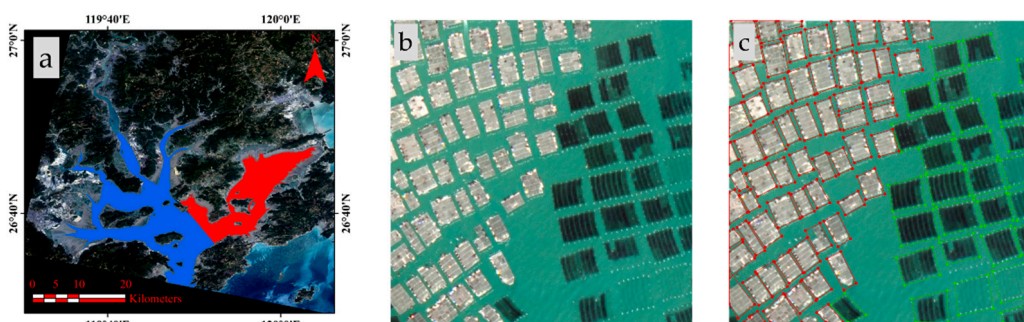

**Figure 7.** Dataset division of Sansha Bay (**a**), where the blue patch represents the range of the training set, and the red patch represents the range of the test set; an example image in the training dataset with a repulsion of 800 × 800 pixels (**b**); building of the training set by labelme (**c**), where the red borders represent cages, and the green borders represent rafts.

According to the research conditions, the model training environment was as follows: Ubnutu 16.04 + intel®Cor (Santa Clara, CA, USA)-eTMi9-10900 XCPU + RTX2060 super + python 3.5 + cuda 10.0 + opencv-python 4.1.1.26 + tensorflow-plot 0.2.0 + tensorflow-gpu 1.13 + tqdm 4.54.0 + shapely 1.7.1 + cpython 0.29.23, and the model training parameters are shown in Table 3.

**Table 3.** Model training parameters.

| Parameter | Value |
|---|---|
| Max epoch | 10 |
| Iteration epoch | 27,000 |
| Max iteration | 270,000 |
| Batch size | 1 |
| Epsilon | 0.00005 |
| Momentum | 0.9 |
| Learning rate | 0.0005 |
| Decay weight | 0.0001 |

### 3.4. $R^3Det$

The R$^3$Det detector [29] is a single-stage detector proposed by Xue et al. by adding a feature refinement module (FRM) to RetinaNet [47]. R$^3$Det is mainly composed of two parts: backbone network and regression subnetwork (classification and bounding box), where the backbone network involves building a feature pyramid network (FPN) [48] on ResNet [49] through top-down paths and horizontal connections. In this way, a rich multiscale feature pyramid is constructed from an input single-resolution image to detect objects at different scales, thereby efficiently extracting features from images. Each layer of the backbone network is connected with a classification and regression sub-network for object classification and location prediction. The horizontal anchor point can achieve a higher recall rate, and the rotation anchor point has a more accurate monitoring effect in dense scenes. Thus, R$^3$Det uses the horizontal anchor point in the first stage to obtain faster speed and higher recall rate, and it uses refined rotation anchors in the refinement stage to detect objects in dense scenes. Meanwhile, to avoid the feature offset caused by the position change of the bounding box, FRM re-encodes the position information of different target bounding boxes to the corresponding feature points, reconstructs the feature map, and realizes the accurate detection of the target. The model structure is shown in Figure 8.

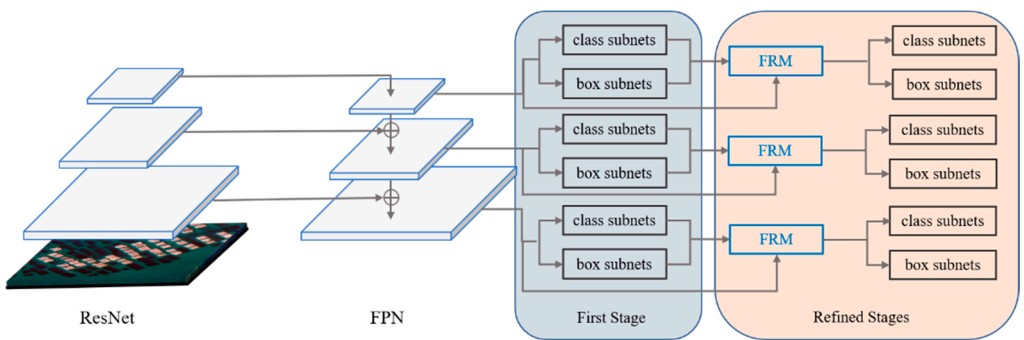

**Figure 8.** R$^3$Det network architecture.

### 3.5. Confusion Matrix

To verify the accuracy of the extraction results in the Sanshawan aquaculture zone. This study adopted the confusion matrix (Table 4) for evaluation. To ensure the generalization and authenticity of the test results, the remote sensing images in the test set were used as the data source to avoid the influence of the training set on the extraction accuracy of the model. On the basis of the extracted result images, this study combined with higher-resolution Google satellite images to visually interpret the extraction targets to obtain relevant aquaculture information, ensuring the accuracy and reliability of the evaluation data [50]. Three commonly used precision evaluation indicators, namely precision, recall, and F-measure, were used to evaluate the extraction precision of the model. Precision and recall indicate the characteristics of a certain classification, while F-measure combines precision and recall and can be used for the overall evaluation of model accuracy. When the F-measure is higher, the classification model is more effective. Therefore, F-measure

was used as the indicator to evaluate the accuracy of the model in extracting aquaculture zones. The specific calculation of the evaluation indicators is as follows:

$$\text{Precision} = \frac{\text{TP}}{\text{TP} + \text{FP}}. \tag{2}$$

$$\text{Recall} = \frac{\text{TP}}{\text{TP} + \text{FN}}. \tag{3}$$

$$\text{F-measure} = \frac{2 \times \text{recall} \times \text{precision}}{\text{recall} + \text{precision}}. \tag{4}$$

**Table 4.** Confusion matrix.

| | | Actual | |
|---|---|---|---|
| | | Positive | Negative |
| **Predict** | Positive | True positive (TP) | False positive (FP) |
| | Negative | False negative (FN) | True negative (TN) |

## 4. Experimental Results and Analysis

### 4.1. Extraction Results

The extraction results of the marine aquaculture zone on the test set based on piecewise linear stretching and R³Det are shown in Figure 9. Raft and cage aquaculture zones are marked with bounding boxes with bright yellow color and dark yellow color, respectively. In addition, "SC" represents the predicted score of cage aquaculture, "RF" represents the predicted score of raft aquaculture, and the score ranges from 0 to 1. A larger score indicates a stronger correlation between the bounding box and the real ground object. The "angle" represents the rotation angle of the bounding box relative to the horizontal.

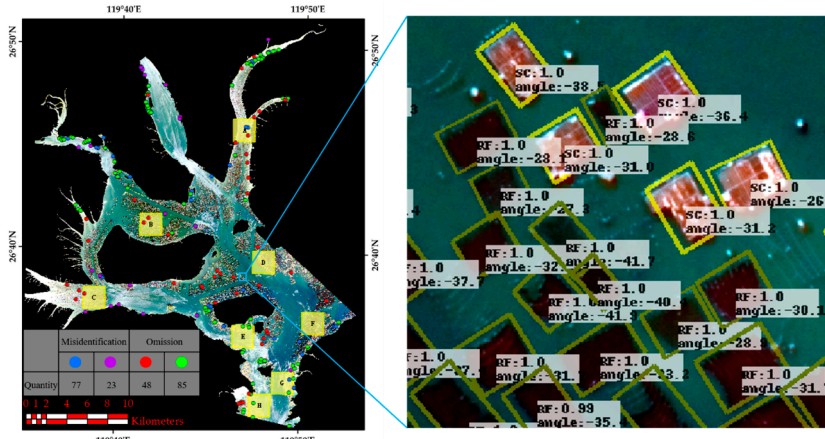

**Figure 9.** Extraction results of aquaculture zones in the test set (**left**) and the enlarged panel showing the extraction details (**right**). The green dot represents missed cages; the purple dot represents misidentified cages; the red dot represents missed rafts; the blue dot represents misidentified rafts.

To intuitively explain the influencing factors of the extraction accuracy, this study examined the missed objects and misidentified objects in the test images. It can be seen from the number and distribution of missed targets and misidentified targets in Figure 9 that the aquaculture zones with poor extraction effect were mainly distributed in coastal zones, especially at the intersection with rivers, and there were many omissions and misidentifications. As for the extraction of the raft aquaculture zone, the number of missed extraction zones was lower than the number of misidentified zones. According to field investigations, it was found that the raft aquaculture zones that were not extracted were mainly small aquaculture zones with inconspicuous appearance characteristics; the zones

mistakenly identified as raft aquaculture were mainly aquaculture cages with low grayscale values because farmers usually cover the top of the cages with a layer of black bird nets to prevent birds from catching fish in the cages. As a result, the grayscale value of the cage aquaculture zone was low, which had a certain impact on the accuracy of raft aquaculture extraction. For the extraction of aquaculture cages, the number of missed extraction cages was much larger than the number of misidentified cages. Misidentified aquaculture cages were mainly affected by ships at sea and were mainly distributed in coastal zones. The missed identification of aquaculture cages was mainly composed of cages with low grayscale values and cages with a small area of the aquaculture zone and oversaturated brightness. The lower grayscale value was mainly due to the existence of the black bird net on the upper layer of the cage, which led to the zone resembling raft aquaculture. Cage aquaculture was similar to raft aquaculture, with many omissions in smaller zones, and this was mainly due to the influence of image resolution, resulting in the loss of target texture features. Moreover, due to the influence of waves, the brightness of the surrounding cages was oversaturated, which led to an increase in the number of missed cages during the extraction process.

### 4.2. Comparisons of Accuracy of Different Stretching Conditions

To verify that piecewise linear stretching could effectively highlight the cage and raft aquaculture zone and improve the image contrast, this study compared piecewise linear stretching with several commonly used image stretching methods, including square root stretching, equalization stretching, Gaussian stretching, and logarithmic stretching. $R^3Det$ was used to classify and extract the aquaculture zones from the stretched images by different stretching methods and the original image, and the comparison results are shown in Table 5. For different stretched images, the F-measures of the extracted cage and raft aquaculture zones by $R^3Det$ were both higher than 90%, and the F-measure of the raft aquaculture zone was higher than that of the cage aquaculture zone. In the extraction results of cages, the F-measure following logarithmic stretching was lower than the unstretched results, and the F-measure following square root stretching and Gaussian stretching was higher than the unstretched results, but the overall improvement effect was not obvious. Furthermore, the F-measure following equalization stretching was higher than the unstretched results, and the F-measure following piecewise linear stretching was the largest, while the effect of extracting cages was the best. Additionally, the recall was smaller than the F-measure. It can be seen that the main factor affecting the accuracy was the missed extraction of cages. In the extraction results of rafts, the F-measures of square root stretching, logarithmic stretching, and Gaussian stretching were all lower than the unstretched results, and the square root stretching and the logarithmic stretching had a lower recall. Furthermore, equalization stretching and piecewise linear stretching had a good effect on the extraction of cultured rafts, and piecewise linear stretching performed the best.

Figure 10 shows the results of $R^3Det$ extracting aquaculture zones under different stretching conditions. To visually show different extraction effects, some annotations are added to the figure, and the changes before and after the annotations are shown in Figure 10g,h. The red rectangles in Figure 10 represent the wrongly extracted aquaculture zones, and the green rectangles represent the aquaculture zones that were not completely enclosed by the bounding box. Compared with the unstretched image (Figure 10f), the overall brightness of the resulting images of square root stretching (Figure 10a) and logarithmic stretching (Figure 10b) was significantly improved, but the contrast between the aquaculture zone and the aquaculture background was not significantly enhanced. Meanwhile, the overall brightness of the resulting image of Gaussian stretching (Figure 10c) was not significantly improved, but the grayscale range of the aquaculture zone was compressed to a certain extent, which reduced the complex characteristics of the aquaculture zone. Although equalization stretching (Figure 10d) enhanced the image brightness and the contrast between the aquaculture zone and the aquaculture background, the noise contrast in the aquaculture background also increased. By contrast, piecewise linear stretching could

not only improve the brightness and contrast of the image but also reduce the complex features caused by different aquaculture periods. According to the number of annotations in Figure 10, under the condition of piecewise linear stretching, R$^3$Det performed better than other stretching methods in extracting the aquaculture zone.

**Table 5.** Comparisons of extraction cage and raft aquaculture zones under different image stretching conditions by R$^3$Det.

| Type | Stretching Method | Precision (%) | Recall (%) | F-Measure (%) |
|------|-------------------|---------------|------------|---------------|
| Cage | Square root stretching | 97.88 | 89.52 | 93.51 |
|      | Logarithmic stretching | 98.57 | 85.09 | 91.33 |
|      | Gaussian stretching | 97.58 | 89.35 | 93.28 |
|      | Equalization stretching | 96.27 | 94.41 | 95.33 |
|      | Piecewise linear stretching | 98.79 | 95.67 | 97.21 |
|      | Unstretched | 98.28 | 88.16 | 92.91 |
| Raft | Square root stretching | 97.17 | 94.31 | 95.72 |
|      | Logarithmic stretching | 97.30 | 90.26 | 93.65 |
|      | Gaussian stretching | 96.66 | 96.41 | 96.53 |
|      | Equalization stretching | 97.13 | 98.61 | 97.86 |
|      | Piecewise linear stretching | 98.66 | 99.16 | 98.91 |
|      | Unstretched | 96.67 | 96.73 | 96.70 |

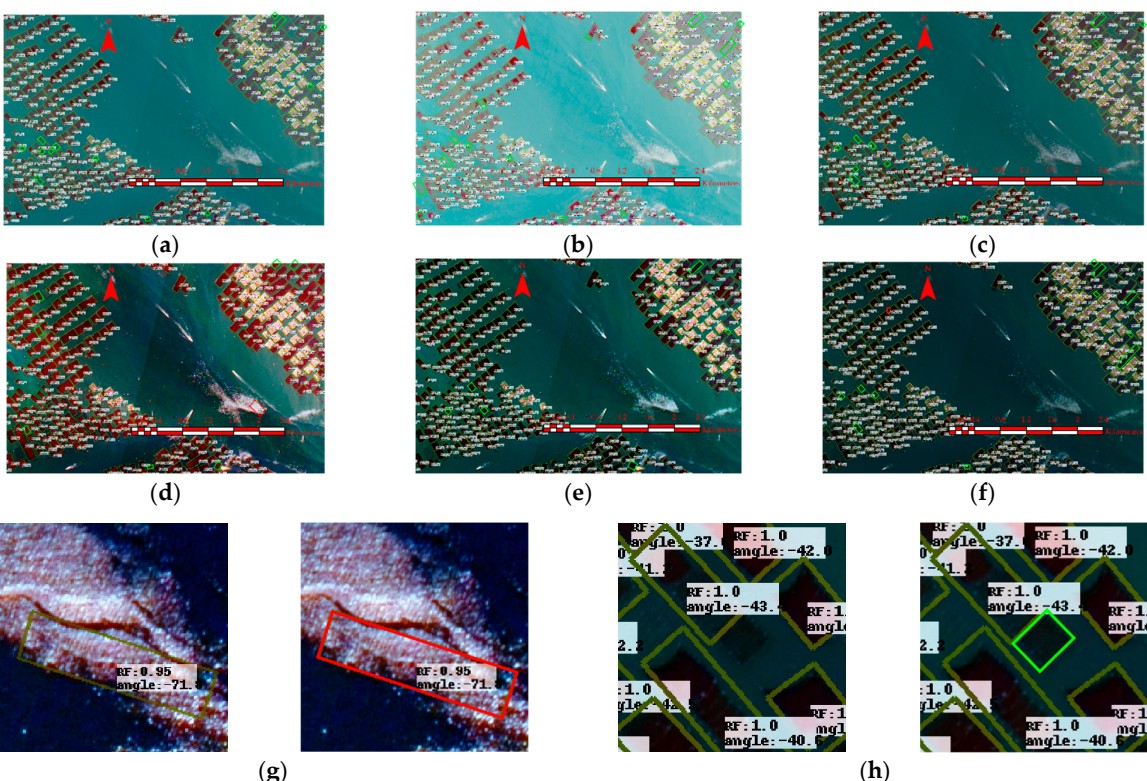

**Figure 10.** Resulting images of R$^3$Det extraction aquaculture zone under different stretching conditions. The red rectangles represent misidentified aquaculture zones, and the green rectangles represent aquaculture zones not included by the bounding box. (**a**) Square root stretching: 13 green rectangles. (**b**) Logarithmic stretching: 14 green rectangles. (**c**) Gaussian stretching: 10 green rectangles and one red rectangle. (**d**) Equalization stretching: seven green rectangles and one red rectangle. (**e**) Piecewise linear stretching (ours): six green rectangles. (**f**) Unstretched: 11 green rectangles and one red rectangle. (**g**) Images of misidentified result (**left**) and manually annotated (**right**). (**h**) Images of aquaculture zones not fully included by the bounding box (**left**) and manually annotated (**right**).

### 4.3. Comparisons of Different Models

Previous research work confirmed the high accuracy and high efficiency of R$^3$Det for aquaculture cages [30]. However, the simultaneous applicability of extracting raft aquaculture and cage aquaculture still requires further verification. R$^2$CNN [51] is a two-stage detector based on the faster R-CNN [52] for detecting text in natural scenes in any direction. It has high accuracy and a high degree of fit between the inclined bounding box and the target, which ensures the advantage of R$^2$CNN in scene text extraction; RetinaNet is a new single-stage detector improved by taking ResNet-101–FPN [48] as the backbone, which can solve the problem of class imbalance by adding a "focal loss" function. Meanwhile, the fit between the rectangular box and the extraction target is improved by adding a rotated rectangular box into RetinaNet [29], which further improves the accuracy of the single-stage detector to extract the target objects from the remote sensing image. In this study, R$^2$CNN, RetinaNet, and R$^3$Det were used to simultaneously extract the raft aquaculture zone and the cage aquaculture zone for comparative analysis. The classification and extraction accuracy of different models for aquaculture zones is shown in Table 6. Although the F-measures of the three models for extracting cage aquaculture and raft aquaculture zones all exceeded 95%, compared with R$^2$CNN and RetinaNet, R$^3$Det had a better extraction effect. Figure 11 shows the partial extraction results of the three models in the marine aquaculture zone under piecewise linear stretching. For the extraction results of R$^3$Det, R$^2$CNN, and RetinaNet, three, 11, and 23 bounding boxes did not fit the aquaculture zone, respectively. Compared with R$^2$CNN and RetinaNet, the bounding box extracted by R$^3$Det fit better with the aquaculture zone. Therefore, the use of R$^3$Det for the statistical analysis of aquaculture zones had higher reliability.

**Table 6.** Comparisons of extraction accuracy of aquaculture zones using different models.

| Type | Model | Precision (%) | Recall (%) | F-Measure (%) |
|------|-------|---------------|------------|---------------|
| Cage | R$^2$CNN | 97.59 | 95.94 | 96.76 |
|      | RetinaNet | 96.97 | 95.87 | 96.42 |
|      | R$^3$Det | 98.79 | 95.67 | 97.21 |
| Raft | R$^2$CNN | 97.82 | 99.10 | 98.45 |
|      | RetinaNet | 96.66 | 98.84 | 97.74 |
|      | R$^3$Det | 98.66 | 99.16 | 98.91 |

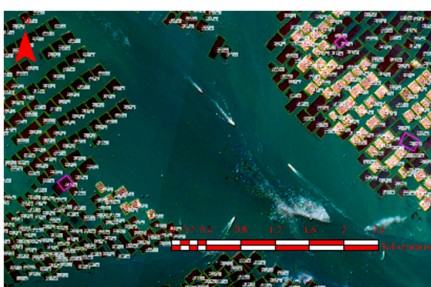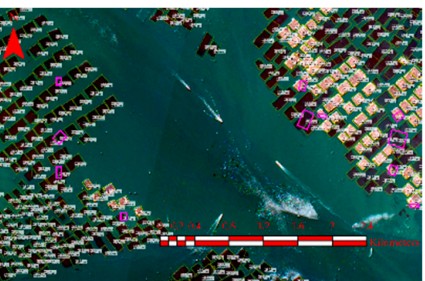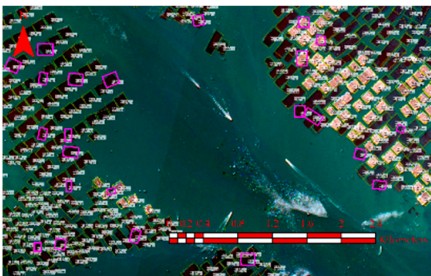

**Figure 11.** Comparisons of the extraction results of R$^3$Det (**left**), R$^2$CNN (**middle**), and RetinaNet (**right**). The pink rectanglew represent bounding boxes that did not fit the aquaculture zones well.

## 5. Discussion

### 5.1. Importance of Piecewise Linear Stretching for Extraction of Aquaculture Zones

The piecewise linear stretching based on image histogram has a good promotion effect on the extraction of coastal aquaculture. However, influenced by factors such as aquaculture types, aquaculture cycles, human intervention, and coastal and estuary sediments, it is difficult to further improve the extraction accuracy of marine aquaculture zones. The piecewise linear stretching takes into account the characteristics of different grayscale

values between the aquaculture zone and the aquaculture background and sets different thresholds between raft aquaculture, non-aquaculture, and cage aquaculture zones, which can reduce the grayscale level of raft aquaculture zones. In this way, the grayscale value of the cage aquaculture and the nearshore sediment deposition zone is improved, and the grayscale value of the raft aquaculture, cage aquaculture, and the nearshore sediment deposition zone is compressed, as shown in Figure 6. Accordingly, the features of raft aquaculture and cage aquaculture zones are highlighted, the contrast of images is improved, and the influence of nearshore and estuary zones on the classification and extraction of marine aquaculture zones is reduced. Compared with the method of using remote sensing technology and improving the network model to improve the extraction accuracy, this method is simpler, more feasible, wider in extraction range, and higher in application value. To further improve the extraction accuracy, traditional remote sensing methods such as object-oriented extraction and threshold segmentation are interfered with by various factors such as sediment, sediment concentration, and chlorophyll concentration in the aquaculture background, making it difficult to effectively distinguish the aquaculture zone from the aquaculture background, thus leading to a high false-positive rate. With the increase in the area, the phenomenon of "same spectrum foreign matter" gradually increases, which also makes it more difficult for traditional methods to extract the target. The method of improving the extraction accuracy of aquaculture zones by improving the network structure of deep learning not only has higher technical requirements for researchers but is also difficult to implement. At this stage, the extraction effect of this method on the marine aquaculture zone has not been significantly improved. According to the extraction results of raft aquaculture and cage aquaculture in Sansha Bay, the precision, recall, and F-measure of $R^3$Det were all greater than 95% under piecewise linear stretching. As a natural aquaculture port, Sansha Bay has an area of 714 square kilometers. There are many estuaries, and the waves at the connection with the outer sea are larger. The successful application of the proposed method in this complex environment shows that the method is not restricted by the area and specific aquaculture environment, and it can provide guidance for the marine management department.

### 5.2. Importance of $R^3$Det for Extraction of Aquaculture Zones

Under piecewise linear stretching, $R^3$Det simultaneously extracts raft aquaculture and cage aquaculture zones with higher accuracy. Moreover, it has higher accuracy and faster speed than the advanced single-stage detectors and two-stage detectors in the field of computer vision. As a more advanced two-stage detector, $R^2$CNN is improved on the basis of fast R-CNN, which not only maintains the extraction accuracy of the two-stage detector but also realizes the tilting of the bounding box. However, the marine aquaculture zone has a large aspect ratio and dense arrangement. There are many "non-fitting" phenomena in the results identified by $R^2$CNN, resulting in more aquaculture background information inside the bounding box. This not only increases the impact of background information on classification accuracy but also reduces the precision of the area extraction. Although RetinaNet introduces a "focal loss" function to solve the problem of class imbalance, the extraction results show (Table 6, Figure 11) that, in the classification and extraction of aquaculture zones, RetinaNet has a lower extraction accuracy and has more "non-fitting" phenomena, resulting in lower reliability when the extraction results are used for statistical analysis in aquaculture zones. By combining the horizontal frame and the rotating frame, $R^3$Det improves the detection speed and accuracy. Furthermore, the feature map can be reconstructed according to the added FRM to achieve feature alignment, ensure the fit of the bounding box and the border of the aquaculture zone, and reduce the influence of non-aquaculture zones.

### 5.3. Influence of the Bounding Box on the Aquaculture Zone

This study verified that piecewise linear stretching and R$^3$Det have good performance in classifying and extracting aquaculture zones. However, the bounding box in the extraction result of R$^3$Det cannot completely fit the actual aquaculture boundary. Therefore, the area of the aquaculture zone is counted according to the coordinate information of the bounding box, and the result obtained is different from the actual aquaculture area. To evaluate the gap between the area obtained by the method proposed in this study and the actual area, eight zones with a size of 4 km$^2$ were randomly selected in the test result image for comparative analysis, and the selected zones are illustrated in Figure 9. In the extraction results, each bounding box had four corresponding coordinate points, allowing the size of the area represented by the bounding box to be obtained. Meanwhile, on the basis of the vectorized data of different types of aquaculture zones in the eight regions, the actual area was obtained.

Figure 12 shows the extraction results and vectorization results of the selected eight regions. It can be seen that the extraction results of the proposed method could effectively avoid the phenomenon of "adhesion" in the aquaculture zone, and the bounding box had good coverage for the aquaculture zone. The detailed results are presented in Table 7. Except for the raft aquaculture zone in zone A, the area of the cage and raft aquaculture zones extracted by R$^3$Det were larger than the vectorized results. This is mainly because the bounding box contained some non-aquaculture zones due to the actual aquaculture zone and the bounding box not fitting completely. In addition, the differences in the extraction precision of the area of cages and rafts in the eight regions were obvious, as the area of the non-aquaculture in the bounding box had a greater contingency for smaller aquaculture zones. Additionally, the precision in the G zone was −189.67%, because the cages in this zone had low grayscale values and were mistakenly identified as raft aquaculture zones. To ensure the validity of the extraction precision, all areas of cages and rafts in the eight regions were evaluated as data sources. The area extraction precision of cage aquaculture and raft aquaculture was 92.48% and 91.88%, respectively, and the extraction precision of the area of the aquaculture zone was 92.08%. Therefore, although the bounding box in the extraction results of R$^3$Det could not completely fit the actual aquaculture boundary, the actual area represented by the bounding box was used for the statistical analysis of the aquaculture area, and the accuracy exceeded 90%, indicating high reliability.

**Table 7.** Extraction precision of the R$^3$Det model for the area of the aquaculture zone.

| ID | Type | Vectorized (Hectare) | R$^3$Det (Hectare) | Precision (%) | Type | Vectorization (Hectare) | R$^3$Det (Hectare) | Precision (%) |
|---|---|---|---|---|---|---|---|---|
| A | Cage | 7.04 | 7.83 | 88.75 | Raft | 74.24 | 66.58 | 89.68 |
| B | Cage | 0.00 | 0.00 | - | Raft | 168.34 | 188.41 | 88.08 |
| C | Cage | 8.98 | 11.53 | 71.60 | Raft | 173.51 | 188.76 | 91.21 |
| D | Cage | 53.30 | 62.38 | 82.97 | Raft | 110.86 | 120.95 | 90.90 |
| E | Cage | 28.80 | 33.83 | 82.53 | Raft | 150.40 | 166.21 | 89.49 |
| F | Cage | 71.31 | 72.06 | 98.95 | Raft | 8.13 | 8.28 | 98.12 |
| G | Cage | 47.97 | 50.59 | 94.53 | Raft | 0.62 | 2.41 | −189.67 |
| H | Cage | 125.49 | 130.45 | 96.05 | Raft | 1.76 | 2.10 | 80.68 |
| A–H | Cage | 342.89 | 368.68 | 92.48 | Raft | 687.85 | 743.70 | 91.88 |

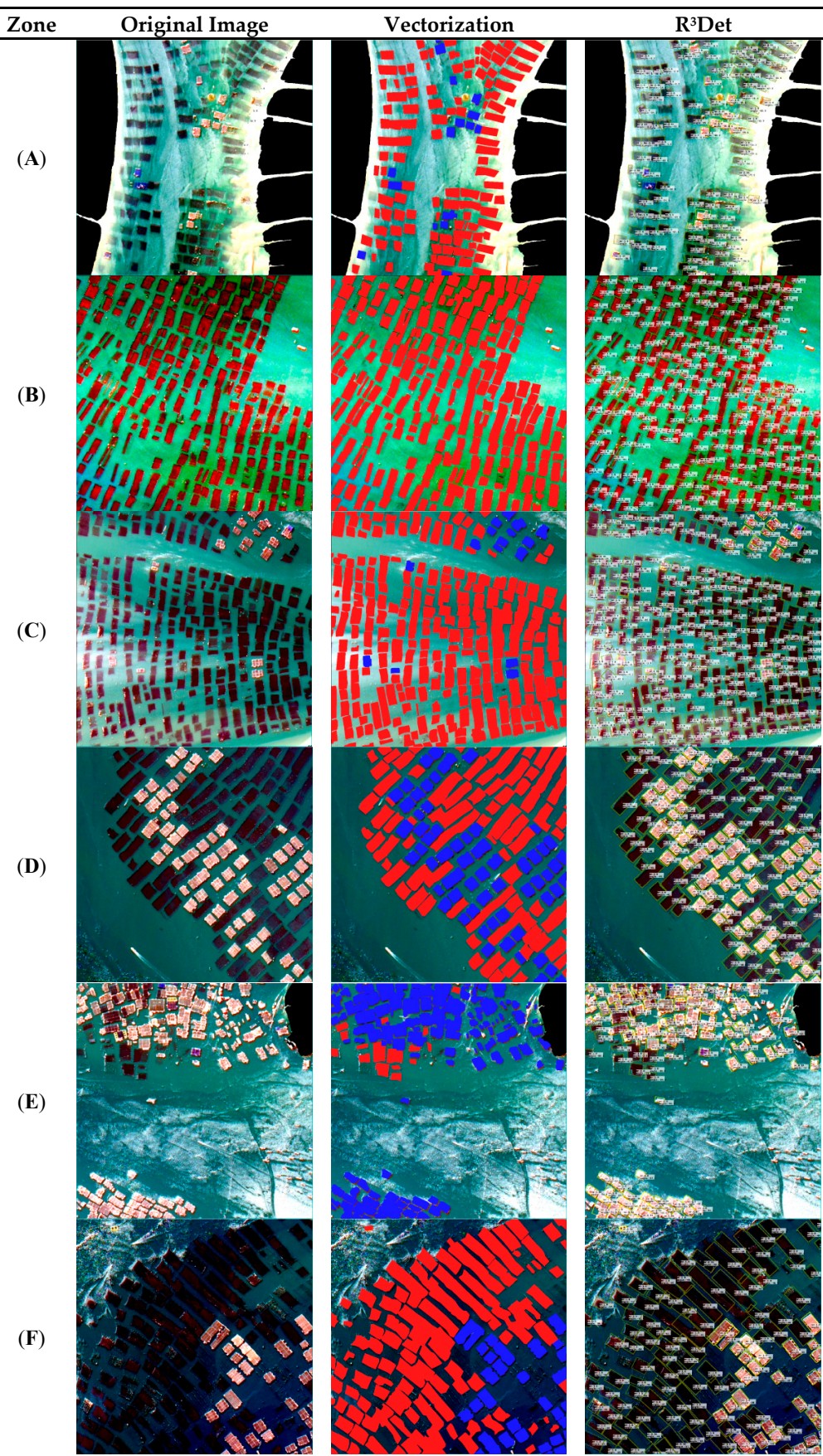

**Figure 12.** *Cont.*

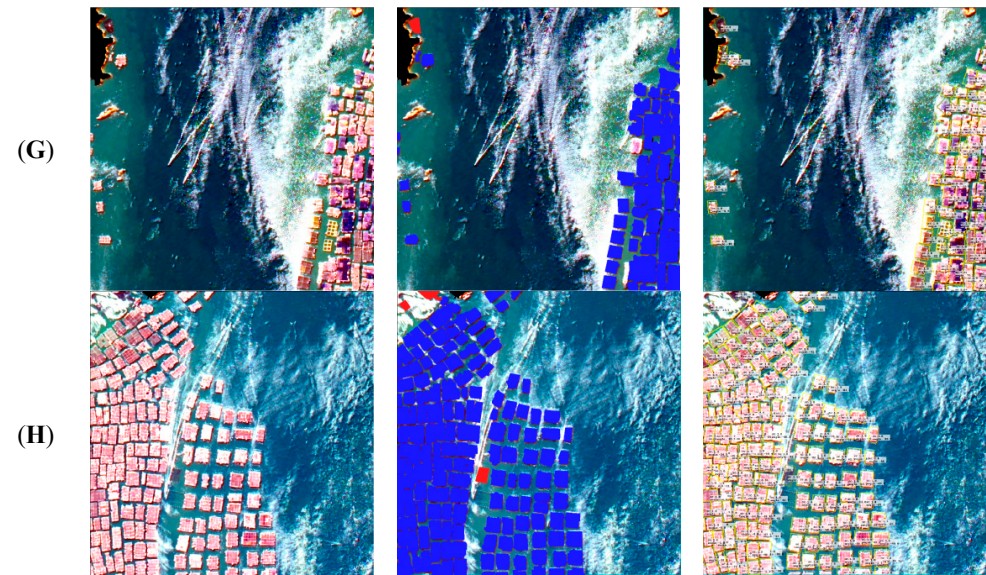

**Figure 12.** Comparisons of extraction precision of the area of the randomly selected aquaculture zones (**A–H**) in Figure 9. The blue patches represent cage aquaculture zones, and the red patches represent raft aquaculture zones.

*5.4. Problems and Prospects*

There are still some shortcomings in the method proposed in this study. To reduce the impact of complex inland on aquaculture zones, when separating land and water, inland water bodies will affect the results, and some aquaculture cages will be eliminated as non-water bodies. This part of the patch needs to be repaired, which inevitably increases the workload. Meanwhile, when performing linear stretching of the image, the threshold setting is artificially determined according to the histogram of the number of pixels with different grayscale scales, and there may be slight differences from the optimal threshold setting, which will affect the image stretching effect and reduce the extraction accuracy. Furthermore, a rectangular bounding box was used to replace the aquaculture zone in the results extracted by R$^3$Det. Although the fit between the bounding box of R$^3$Det and the object was the highest in the rotating object detection model, the result was still affected by non-aquaculture zones.

In future work, we will try to extract the boundary of the aquaculture zone according to the grayscale difference of the object in the bounding box, establish the area of interest of the aquaculture zone, and further improve the extraction precision of the aquaculture area. With the improvement of multispectral satellite resolution, the development of the aquaculture industry can progress more effectively by establishing a more reasonable aquaculture plan via the extraction of aquaculture information from the remoted images. In addition, marine litter is an important factor affecting the marine environment, and efficient removal of marine litter is of great significance to the protection of marine ecology [53,54]. In the future, we will investigate the performance of the proposed method in identifying marine litter.

## 6. Conclusions

This study took the aquaculture zone of Sansha Bay in 2020 as the research area and proposed a new method for marine aquaculture zone extraction. This method highlighted the features of the aquaculture zone through piecewise linear stretching, which further improved the classification and extraction accuracy of R$^3$Det for marine aquaculture zones. The conclusions are as follows:

1.  Compared with the stretched images using methods of square root stretching, equalization stretching, Gaussian stretching, logarithmic stretching, and unstretched images, piecewise linear stretching could more effectively highlight the appearance characteristics of raft aquaculture and cage aquaculture zones, as well as improve the contrast of the images, achieving the highest accuracy for both raft and cage extraction.
2.  Compared with $R^2$CNN and RetinaNet, $R^3$Det showdc a higher extraction accuracy for marine aquaculture zones under piecewise linear stretching. The overall extraction accuracy of $R^3$Det for Sansha Bay raft aquaculture and cage aquaculture were 98.91% and 97.21%, respectively, and the extraction precision of the total area of aquaculture was 92.08%.
3.  The method proposed in this study is not limited by factors such as specific aquaculture zones and model structure and can classify and extract marine aquaculture zones under large-scale and complex aquaculture backgrounds. The study results can provide effective assistance for relevant marine aquaculture management departments to conduct large-scale aquaculture monitoring and scientific sea use, thus achieving sustainable development of the marine aquaculture industry.

**Author Contributions:** Conceptualization, Y.M. and X.Q.; methodology, Y.M.; software, X.Q.; validation, P.Z. and H.H.; investigation, F.G.; resources, D.F.; data curation, D.F.; writing—original draft preparation, Y.M. and X.Q.; writing—review and editing, C.Y., L.W., F.G. and D.F.; visualization, Y.M. All authors have read and agreed to the published version of the manuscript.

**Funding:** This research was funded by the National Key Research and Development Projects (2020YFE0200100) and the National Natural Science Foundation of China (42076213).

**Acknowledgments:** The authors would like to thank all reviewers and editors for their comments on this paper.

**Conflicts of Interest:** The authors declare no conflict of interest.

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
