# Peer review of "Automatic Extraction of Marine Aquaculture Zones from Optical Satellite Images by R3Det with Piecewise Linear Stretching"

_remotesensing, doi:10.3390/rs14184430_

Round 1
Reviewer 1 Report
Dear authors,
The effort to evaluate the potential offered by remote sensing data, machine learning and cloud computing to classify the changes that occurred in fire-affected areas is interesting and would bring a significant contribution in this field.
Before this paper could be considered for publication, there are a few concerns that need to be addressed.
L58-L64. In my opinion it’s better to add First name of the cited author, because it’s hard to understand what does covered under “Seto”, “Jayanthi”, etc. It should be changed to “K.C. Seto …” and “M. Jayanthi …”
L152. In my opinion, at the first mention of any abbreviation it is better to give full name of the satellite – Gaofen-6 (GF-6)
L180. What was the reason for the use of NDWI, and not, for example, manual digitization. On the one hand, this would increase the time for preparing a data set, on the other hand, it would increase the accuracy of labeling, which would lead to more accurate results.
L326. In my opinion the labels are too small to be able to see anything on them. Their importance in this case is not significant. Perhaps it should be removed for a clearer visualization of bounding-boxes
I wish that my comment would be helpful in improving the quality of this research.
Thank you.
Reviewer 2 Report
This work addresses an important topic concerning the extraction and classification of aquaculture zones from the satellite images. This study proposed a method based on the combination of piecewise linear stretching and R3Det to classify and extract raft aquaculture and cage aquaculture zones. This represents an important step-forward in the research topics of deep learning for the satellite image classification.
I have only few comments to improve this work.
In the subsection 3.1 Extraction process, some sentences concerning the assessment of NDWI (explain the acronym) could be inserted, in order to understand how the features of water bodies can be detected inland. Furthermore, how did you assess the NDWI from panchromatic images of GF-6?
In the subsection 3.3 Dataset, the hardware characteristics of the workstation used for the training of the artificial intelligence must be inserted, because they are important for the accuracy assessment.
The section 5. Discussion lacks reference, there are different works that could be compared for the classification methods (see 10.1016/j.envpol.2021.116490; 10.1088/1748-9326/abbd01).
Other comments are highlighted in the attached pdf file.
Many thanks and kind regards.

Reviewer 3 Report
The authors analyzed the GF-6 satellite observations to identify the aquaculture area in the coast of Fujian Province. The result shows prominent spatial pattern and the rafts are clearly identified. The study can be applicable to other areas and could offer important insight for assessing the impact of aquaculture on regional dynamics and ecosystem in the marginal seas. Some minor changes are suggested to further improve the study.
1. What is the spatial domain of GF-6 observations? The method sections described the width of band is 95km. And is the satellite cover the entire coastal area of China? Some information describes the spatial range will be highly valuable.
2. How long is the high-precise observations? Besides GF-6, the other satellite can be involved to expand the spatial and temporal coverage.
3. Is the satellite observations also applicable for identifying the other parameters in the aquaculture region? For example, the satellite observed chlorophyll or water quality.
Reviewer 4 Report
In this paper, the marine aquaculture is stretched by piecewise linear stretching, and the raft aquaculture target and cage aquaculture target in Sansha Bay are extracted based on R3Det network. The effectiveness of the method and model is verified through comparative experiments. The structure of the paper is complete and the logic is clear, but the following problems need to be modified:
1. The expression effect of some figure is not ideal.
The text in Fig. 1 should be enlarged. The scale does not need to be displayed too much by level. At the same time, the scale value is preferably a multiple of 10. The scale bar in Fig. 7 has the same problem. The target detection box of most pictures is not obvious, and a more obvious color should be selected.
2. The deep learning network can only input three channels of data, but it is not necessary to input according to the actual channel of RGB. The NIR band in the remote sensing image may contain rich information. It is better to verify the difference between the aquaculture target and the non-aquaculture zone in different bands, and select the 3 bands with the largest difference in information to input the network for training, instead of directly removing the NIR band.
3. The details of parameter setting during R3Det network training shall be supplemented.
4. Figure 6 shows the comparison before and after the piecewise linear stretching process. Currently, the display effect is poor and should focus on the extracted raft and cage targets.
5. The expression effect of Figure 10 is not satisfactory. The extraction target should be further enlarged to find typical areas to highlight the difference between the extraction results of the article and those of other methods.
